# Effect of Omega-3 Microcapsules Addition on the Profile of Volatile Compounds in Enriched Dry-Cured and Cooked Sausages

**DOI:** 10.3390/foods9111683

**Published:** 2020-11-18

**Authors:** Juan Carlos Solomando, Teresa Antequera, Alberto Martín, Trinidad Perez-Palacios

**Affiliations:** 1Research Institute of Meat and Meat Products (IProCar), University of Extremadura, Avda. de las Ciencias s/n, 10003 Cáceres, Spain; juancarlosg@unex.es (J.C.S.); tantero@unex.es (T.A.); 2Nutrition and Food Science, Institute of Agri-Food Resources (INURA), University of Extremadura, Avda. Adolfo Suárez s/n, 06007 Badajoz, Spain; amartinunexes@gmail.com

**Keywords:** volatile compounds, fish oil microcapsule, dry-cured sausage, cooked sausage

## Abstract

The main goal of the present study was evaluating the effect of enriching meat products (cooked (C-SAU) and dry-cured sausages (D-SAU)) with monolayered (Mo) and multilayered (Mu) fish oil microcapsules on the profile of volatile compounds, with special interest in lipid oxidation markers. For that, Solid-Phase Microextraction (SPME) and Gas Chromatography-Mass Spectrometry (GC-MS) were used. Significant differences were found in the volatile compound profile between Mo and Mu, which was been reflected in the meat samples. Thus, in general, volatile compounds from lipid oxidation have shown higher abundance in Mo and C-SAU and D-SAU enriched with this type of microcapsule, indicating that the wall of Mu (chitosan-maltodextrine) might protect the encapsulated bioactive compounds more effectively than that of Mo (maltodextrine). However, this finding is not reflected in the results of previous studies evaluating the sensory perception and oxidation stability of C-SAU and D-SAU, but it should be considered since unhealthy oxidation products can be formed in the enriched meat products with Mo. Thus, the addition of Mu as an omega-3 vehicle for enriching meat products may be indicated.

## 1. Introduction

Numerous epidemiological studies suggest that a diet rich in omega-3 polyunsaturated fatty acids (ω-3 PUFAs), mainly eicosapentaenoic acid (EPA; C20:5 ω-3) and docosahexaenoic acid (DHA; C22:6 ω-3), has a marked influence in the avoidance and therapy of a series of chronic disorders [1], particularly coronary heart disease [2,3,4,5]. However, Western diets are poor in long chain ω-3 PUFAs and the liking of fish and seafood products (main sources of EPA and DHA) is currently static or declining [6]. Therefore, to improve the welfare state of the population, different professional organizations and health agencies have recommended the consumption of around 0.25 g EPA + DHA per person and day [7,8,9,10]. There also is a European Union law that sets up the minimum quantities of EPA + DHA (40 and 80 mg per 100 g and per 100 kcal) to label a food as a “source of ω-3 fatty acids” and “high in ω-3 fatty acids”, respectively [11].

Nowadays, meat products are highly appreciated, with a habitual consumption of around 3–4 times per week [12], which is principally ascribed to the modern standards of life and the demand of “ready-to-eat” products. The content of high biological value proteins in meat products is valued; however, the percentage saturated fatty acids (SFA) and the ratio of ω-6/ω-3 PUFA is not so desirable in some of these foods [13]. Hence, industries are focused on the developing of meat products with a healthier lipid profile [14], there being some investigations on the possibility of incorporating fish oils, as bulk or emulsified, in different foods [15,16,17]. However, the presence of numerous double bonds in ω-3 PUFAs causes a rapid oxidation in contact with oxidant promoters, like iron, light, oxygen and high temperatures [18,19] that accelerate the formation of primary oxidation products, such as hydroperoxides, which easily isomerize and degrade to volatile compounds. Some of them impart undesirable off-fishy and rancid odors and flavors such as, 4-heptenal, 3,5-octadiene or 2-ethylfuran [20,21,22].

In this context, several studies have investigated the possibility of producing stable foods enriched with ω-3 PUFA microcapsules [15,23,24,25]. The microencapsulation technique aims to form a wall around the active compounds, reducing the perception of off-flavors [26] and the contact with oxidant promoters [18,19]. Moreover, this technique is of easy application in the food industry, economic and scalable [27]. Recent studies have pointed out the possibility of adding fish oil microcapsules to enrich different meat products, which have been recently reviewed [28,29].

Most studies on enrichment of meat products with ω-3 PUFAs microcapsules have focused on the evaluation of the proximal composition, oxidative stability, fatty acid profile and sensory attributes of the enriched foods [24,25,30,31], without taking into account the influence of the different processing conditions and microcapsules addition on the volatile compounds profile of these meat derivatives, except for a previous study in chicken nuggets [32].

Authors of the present study have developed two types of fish oil microcapsules (from homogenized monolayered (Mo) and multilayered (Mu) fish oil emulsions) with high quality characteristics in terms of yield, microencapsulation efficiency and oxidative stability [29]. Results on meat products have shown similar quality characteristics in control and enriched batches with Mo and Mu fish oil microcapsules [28].

The main objective of the present study was to evaluate the profile of volatile compounds of cooked and dry cured meat products enriched with monolayered and multilayered fish oil microcapsules, paying special attention to those from oxidation processes. The profile of volatile compounds of the fish oil microcapsules was also investigated.

## 2. Materials and Methods

### 2.1. Experimental Design

This study was carried out with cooked (C-SAU) and dry-cured sausages (D-SAU). For each meat product, three batches were prepared: added with monolayered (Mo) (C-SAU-Mo and D-SAU-Mo) and multilayered (Mu) microcapsules (C-SAU-Mu and D-SAU-Mu), and a control batch (without enriching) (C-SAU-Co, D-SAU-Co). In the added bathes, the batter was modified by the addition of 2.75% (*w/w*) of Mo and 5.26% (*w/w*) of Mu. These are 3 and 5 g of Mo and Mu, respectively, per 100 g of dough. The different quantities of Mo and Mu added are due to the differences between Mo and Mu in the efficiency of fish oil encapsulation (87.39 and 56.43%, respectively) and consequently, in the quantity of EPA and DHA (2.75 and 5.26 g EPA + DHA per 100 g of microcapsule, respectively). The quantities of Mo and Mu were deliberately added in order to give 40 mg of EPA + DHA per 100 g of meat products and labelling with “source of ω-3 fatty acids”.

The profile of volatile compounds was analyzed in the three batches of C-SAU and D-SAU, and also in Mo and Mu.

### 2.2. Preparation of Omega-3 Sources

The source of EPA and DHA was fish oil from cod liver, supplied by Biomega Nutrition (Silkeborg, Spain) and containing 5.96% EPA and 25.83% DHA. Mo and Mu microcapsules were prepared following the procedure described by [33], with slight modifications. Firstly, Mo and Mu emulsions were prepared, by mixing fish oil (20 g) and soybean lecithin (6 g), provided by Across Organics (Madrid, Spain), with a magnetic stirring overnight. Water was added until the mixture reached a total weight of 200 g. The mixture was homogenized (20,000 rpm, 10 min) using an Ultraturrax T-18 basic (IKA, Königswinter, Germany), with a primary emulsion that was homogenized at high-pressure (SPX, model APV-200a, Silkeborg, Denmark). The high-pressure homogenization conditions have been previously optimized: 1200 Ba and 3 passes for Mo and 1100 and 2 passes for Mu [29]. The next step was different for each type of emulsion: water (200 g) was blended with Mo, while 200 g of 1% of chitosan (*w/w*) with 95% of deacetylation (Chitoclear FG 95, kindly provided by Trades, Murcia, Spain) in acetic acid 1% were added to Mu. The mixtures were slowly agitated with a magnetic stirrer for 15 min. Finally, in Mo and Mu, 400 g of maltodextrin solution (120 g maltodextrin + 280 g water) with a dextrose equivalent of 12% (Glucidex 12, kindly provided Roquette, Lestrem, France) were added. Thus, feed emulsions (800 g) of Mo and Mu were obtained.

A laboratory-scale spray drier equipped with a 0.5-mm nozzle atomizer (Mini spray-dryer B-290, Buchi, Switzerland) was used to dry the feed emulsions, applying the following parameters (80% aspirator rate = 80%, feed rate = 1 L/h, inlet temperature = 180 °C, outlet temperature = 85–90 °C). During the spray-drying process, the emulsions were agitated in a magnetic stirrer at room temperature. The dried powders collected were refrigerated (4 °C) until they were added to the meat products. Quality characteristics of Mo and Mu emulsions and microcapsules have been previously analyzed [29].

### 2.3. Elaboration of Meat Products 

The ingredients for the elaboration of C-SAU were meat mechanically separated from chicken (60%), water (20%), pork fat (12%), salt (14g/kg), pork plasma (8%), stabilizer (E-450), aromas, vegetable fiber, spices, spice extracts, smoke flavor, antioxidant (E-316) and preservative (E-250). Moreover, the enriched batches were added with the corresponding microcapsules in the knead phase. The mincer Asgo MVZ 600 (Oporto, Portugal) was used for mixing. All C-SAU batches followed the same processing: stuffing in 18 mm diameter cellulose casings, using a Stuffer Vernag HPE (Barcelona, Spain), heating at 85 °C for 30 min in a cooking pot Gaser MCA 200 (Girona, Spain), cooling at 7 °C for 1 h, and finally being vacuum packed. Samples were maintained in refrigeration (0–5 °C for 7–9 days), heated (90 °C, 3 min) and analyzed. 

In the case of D-SAU, the ingredients were Iberian pork meat (80%) and fat (15%), salt (20 g/kg), dextrose, soy protein, spices, aromas, stabilizers (E-451 and E-450), antioxidant (E-301), preservatives (E-252 and E-250), flavor enhancer (E-621), coloring (E-120) and the corresponding microcapsules in the case of the enriched batches. The meat and the fat were minced (Sheydelman AU 200, Aalen, Germany), through a 6 mm diameter mincing plate, mixed with the rest of ingredients (Asgo MVZ 600, Porto, Portugal). No starter culture was added. Collagen casings (40 cm length, 60 mm diameter) and a Stuffer Vernag HPE (Barcelona, Spain) were used to stuff the dough. Once stuffing, the products were dry-cured processed in three consecutive stages: (i) 4 °C, 82% of relative humidity, 3 days; (ii) 8 °C, 80% of relative humidity, 21 days; (iii) 5 °C, 85% relative humidity, 14 days. At the end of this process, the percentage of loss was 38–40%. The dry-cured sausages were analyzed after storing at ambient temperature (18–20 °C for 7–9 days). 

C-SAU and D-SAU batches were formulated and manufactured in a meat industry (remain anonymous). 

### 2.4. Analysis of Volatile Compounds

Microcapsules and minced meat products were sampled (1 g) into a 10 mL glass flask (Hewlett–Packard, Palo Alto, CA, USA) sealed with an aluminum cap and polytetrafluoroethylene (PTFE) butyl septum (Perkin-Elmer, Foster City, CA, USA). A solid-phase microextraction (SPME) method [34] was first applied for the volatile compound extraction. A cross-linked carboxen/polydimethylsiloxane fiber (10 mm long, 100 µm thick; Supelco, Bellefonte, PA, USA) was used. It was conditioned (220 °C, 50 min) in the gas chromatograph (GC) injection port before use. For the absorption of the volatile compounds of samples, the fiber was introduced in the sealed vial, being placed in a water bath a 40 °C, for 30 min. After that, SPME fiber was moved to the injection port and maintained for 30 min for desorption. Analyses were performed using a Hewlett–Packard 6890 series II GC coupled to a mass selective detector (HP 5973) (Hewlett–Packard, Wilmington, DE, USA). A 5% phenyl–95% polydimethylsiloxane column (30 m × 0.32 mm ID, 1.05 μm film thickness, Hewlett–Packard) was used, operating at 40 °C, with a column head pressure of 6 psi of and a flow of 1.3 mL min^−1^. The mode of the injection port was splitless. The following temperature program was applied: isothermal for 15 min at 35 °C, increased to 150 °C at 4 °C min^−1^, and then to 250 °C at 20 °C min^−1^. Temperature of the transfer line to the mass spectrometer was 280 °C. The mass spectra were obtained using a mass selective detector by electronic impact at 70 eV, a multiplier voltage of 1756 V and collecting data at a rate of one scan over the m/z range of 30–550 u.m.a. The linear retention indexes (LRIs) of the volatile compounds were calculated by means of N-alkanes (Sigma R-8769), which were analyzed under the same conditions. Identification of volatile compounds of samples was done by comparison of mass spectra with databases (National Institute of Standards and Technology (NIST) and Wiley libraries) and by comparison of their LRI with those available in the literature [20,21,35,36,37,38,39,40,41,42,43,44,45,46,47,48,49,50,51,52,53,54,55,56,57,58]. Results from volatile analyses are provided in area units (AU).

### 2.5. Sampling Replication and Statistical Analysis

Replicate experimental samples (*n* = 3) of Mo and Mu microcapsules and of the three batches (Co, Mo and Mu) of meat products (C-SAU and D-SAU) were analyzed in triplicate. One-way analysis of variance (ANOVA) was applied to evaluate the addition of different types of fish oil microcapsules and the differences between microcapsules. In the case of being significant effects (*p* < 0.05), paired comparisons between means were conducted using Tukey’s test. Furthermore, all volatile compounds that showed significant differences in the ANOVA analysis were included into a principal component analysis (PCA). The statistics were run using the program IBM SPSS Statistics v.22 (IBM, Armonk, NY, USA).

## 3. Results and Discussion 

### 3.1. Volatile Compounds in Fish Oil Microcapsules

A total of 40 volatile compounds were identified in the Mo and Mu fish oil microcapsules, which were grouped in the following chemical families: aliphatic hydrocarbons, alcohols, aldehydes, ketones, furans and acids. Figure 1 shows the area percentage of each chemical family in Mo and Mu. In both types of fish oil microcapsules, aliphatic hydrocarbon was the major chemical family, followed in decreasing order by aldehydes, ketones, alcohols, furans and acids. This profile of volatile compounds is quite according to a previous study with double and multilayered fish oil microcapsules [35]. The percentage of aliphatic hydrocarbons has been used as an indicator of quality and stability of different commercial fish oils, from salmon, tuna, sardines and shrimp. Considering the relationship between the decrease in the percentage of this family of volatile compounds with an increase in lipid oxidation [51], the high percentage of aliphatic hydrocarbons in Mo and Mu may support the protective effect of the wall materials of these types of microcapsules, minimizing the contact and reactivity of fish oil with oxidizing promoters. Significant differences were detected in the percentage of the most chemical families of volatile compounds between the Mo and Mu of this study, showing Mo higher percentages of aldehydes (8.43% vs. 4.78%), ketones (3.82% vs. 2.65%), alcohols (1.97% vs. 1.10%) and lower of aliphatic hydrocarbon (87.89 vs. 90.43%) and acids (n.d. vs. 0.32%) than Mu. Accordingly, [59] also found significant differences in the percentage of chemical families of volatile compounds between different types of fish oil microcapsules.

Table 1 lists the individual volatile compounds of Mo and Mu, being expressed as AU × 10^6^. Hexane was the major volatile compound in Mo and Mu, followed by pentane, hexanal and 3-hydroxy-2-butanone, and the rest of individual volatile compounds showed less than 1 AU × 10^6^. This agrees with results described for double and multilayered fish oil microcapsules [35]. From the 40 individual volatile compounds identified in the fish oil microcapsules of the present study, 9 of them were found in Mu but not in Mo (tridecane, 1,2,4-butanetriol, phenylethyl alcohol, 2-heptenal, 2-octenal, 2-nonanone, 2-butyltetrahydrofuran, heptanoic and sorbic acid) and 7 were only detected in Mo (1-heptene, heptane, decane, 2-propanol, 4-hexen-1-ol, 1-heptanol and 2-hexenal). 

Significant differences were found in most individual volatile compounds between Mo and Mu (Table 1). Regarding the aldehydes, which have been described as the most important indicators of fish oil oxidation [51], Mo showed significant higher levels of propanal, pentanal, hexanal, 2-hexenal, heptanal, octanal and nonanal compared to Mu. It has been described that propanal comes from the lipid hydroperoxides derived from ω-3 PUFA while hydroperoxides derived from ω-6 PUFA mainly generate hexanal, as consequence of the breakdown of the first double bond of the n position of the ω-3 and ω-6 fatty acids, respectively [60]. In addition, hexanal has been used in previous studies as a marker to measure the quality and oxidative stability of fish oil microcapsules [20]. On the other hand, Mu had a significant higher AU of 2-pentenal and 2-octenal than Mo, but these two volatile compounds have not been related to the oxidation of ω-3 PUFAs. In addition, other relevant indicators of fish oil oxidation, such as 2,4-heptadienal and 2,4-decadienal, which have been associated to the perception of rancid flavor [20,48,60], or other aldehyde volatile compounds related to rancid odors, such as decanal or 2-nonenal [20,48,61], have not been identified in Mo or Mu.

Most individual alcohols and ketones have shown significant differences between Mo and Mu, with higher AU in Mo in comparison to Mu in most cases. However, as is our knowledge, either of the ketones detected in Mo and Mu are associated with the lipid oxidation process. The most important ketones from lipid autoxidation reactions are 3,5-octadien-2-one and 1-octen-3-one [49]. In fact, they have been detected in mayonnaise and milk enriched with fish oil, being strongly correlated with the strength of the oxidation process [48,50]. However, these ketones were not detected in Mo or Mu. Regarding the alcohols, 1-penten-3-ol and 2-penten-1-ol have been described as one of the most characteristics oxidation markers for PUFA [48,49]. 1-penten-3-ol was detected in both types of microcapsules, with higher AU in Mo than in Mu, while 2-penten-1-ol was not found in these fish oil microcapsules. Another common oxidation product of ω-3 PUFA is 2-ethylfuran, which can be generated from the 12-hydroperoxide of EPA and 16-hydroperoxide of DHA [35,62]. This volatile compound has been identified in both types of microcapsules, Mo having significantly higher AU than Mu. Two volatiles compounds, heptanoic and sorbic acid, were found in Mu, but not in Mo. Heptanoic acid at high concentrations imparts unpleasant rancid odor, but the AU of this compound in Mu are very low.

The higher AU in some individual volatile compounds related to lipid oxidation found in Mo in comparison to Mu could be explained by the different wall material of these fish oil microcapsules, being of maltodextrine and of chitosan plus maltodextrine, respectively. In fact, it has been described that chitosan increases the electrostatic force and viscosity of the layers [63], avoids the oxidative damage and could act as a free scavenger [64]. Thus, the Mu coating may be more effective than Mo to protect fish oil from oxidative damage. This aspect can be marked in the case of volatile compounds such as hexanal, 1-penten-3-ol and 2-ethylfuran, with low odor thresholds (4.5, 1 and 2.2 µg kg^−1^ oil, respectively) and associated with sensory defects [65,66].

Nevertheless, in comparison to the profile of volatile compounds in bulk fish oil, Mo and Mu have not shown to be polyunsaturated lipid oxidation products with rancid taste perceptions, such as 2,4-heptadienal, 2,4-decadienal, 2-nonenal, 3,5-octadien-2-one and 1-octen-3-one [51,67], which points out the effectiveness of the Mo and Mu microcapsules of the present study to minimize the contact and reactivity of fish oil encapsulated with oxidizing promoters. 

### 3.2. Volatile Compounds in Dry-Cured and Cooked Sausages Enriched with Fish Oil Microcapsules

A total of 53 and 60 volatile compounds were identified in D-SAU and C-SAU of the present study, respectively, which were grouped in the following chemical families: aliphatic hydrocarbons, alcohols, aldehydes, furans, ketones, terpenes, acids, esters, aromatics, cyclic hydrocarbons and pyrazines. Figure 2A,B show the percentages of these chemical families of volatile compounds in D-SAU and C-SAU as affected by type of fish oil microcapsule addition, respectively. The most abundant families in all batches of D-SAU were acids and aldehydes, followed far behind by terpenes and esters. Minor percentages were found for aliphatic hydrocarbons, aromatics, ketones, cyclic hydrocarbons, alcohols and furans. This profile is quite in concordance with previous studies in fermented sausages [53,58]. Moreover, significant differences were detected in the chemical families of volatile compounds between the D-SAU batches of this study, with Mo showing higher percentages of aldehydes and terpenes (16.03 and 8.2) than Co and Mu batches, and Mu having a higher percentage of acids (69.63) and lower percentages of esters (5.58) than Co and Mo. 

In C-SAU, the major family of volatile compounds was aldehydes, followed by aliphatic hydrocarbons, cyclic hydrocarbons and alcohols, while minor abundance was detected for terpenes, acids, esters, aromatics, ketones, furans and pyrazines. This agrees with the profile of volatile compounds previously reported in other studies in cooked sausages [42,46]. Moreover, significant differences were found in the chemical families of volatile compounds between the C-SAU batches of this study, showing in C-SAU-Mo higher percentages of aldehydes and alcohols than in C-SAU-Co and C-SAU-Mu; C-SAU-Mu obtained a higher percentage of acids and esters and a lower percentage of aliphatic hydrocarbons than C-SAU-Co and C-SAU-Mo, and the percentage of cyclic hydrocarbons was higher in C-SAU-Co than in the enriched batches. Thus, at first, considering the results on the percentage of volatile compounds, the differences between microcapsules are reflected in the bathes of meat products. Taking a step forward, the individual volatile compounds of the control and enriched batches of the meat products of the present study are analyzed in the following sections.

Table 2 lists the individual volatile compounds of D-SAU as affected by the type of fish oil microcapsule added. Acetic acid was the major volatile compound in all batches, followed by hexanal, methyl hexanoate, β-myrcene, pentanoic acid, butanoic acid and heptanal, the rest of the individual volatile compounds showing less than 10 AU × 10^6^. This profile is quite in concordance with previous studies in dry fermented sausages [53,55]. The high content of acetic acid would be related to microbial fermentation of carbohydrates [55,68]. Others compounds also typical of carbohydrate fermentation, such as 3-hydroxy-2-butanone, were also detected in D-SAU batches [57]. The high AU of β-myrcene in these samples is also noted, which may be ascribed to the addition of species [69].

The enrichment effect with Mo and Mu fish oil microcapsules significantly influence the volatile compounds of most chemical groups (Table 2), excluding esters, aromatics and cyclic hydrocarbons. Only 13 in 53 volatile compounds identified in D-SAU showed significant differences among batches: C-SAU-Mo showed higher AU in six volatile compounds (1-propanone, 1-penten-3-ol, 1-octen-3-ol, pentanal, 3,5-octadien-2-one and acetic acid) and lower in one (butanal) in comparison to C-SAU-Co and C-SAU-Mu; C-SAU-Mu obtained higher abundance in two volatile compounds (heptane and β-mycene) and lower in one (3-hydroxy-2-butanone) than in C-SAU-Co and C-SAU-Mo, and C-SAU-Co showed higher AU in one volatile compound (2-penthyl-furan) and lower in two (1-propanol and 2-ethylfuran) in comparison to the enriched batches. Figure 3a represents the score plots of the PCA of volatile compounds data from the D-SAU samples. The first principal component (PC1) comprised 55.48% of the total variance, and the second principal component (PC2) accounted for 29.47%. The score plot indicates a clear differentiation of samples as affected by the addition of fish oil microcapsules: those with high positive PC1 scores (D-SAU-Mo), those with high positive PC2 scores (D-SAU-Mu) and those with high negative PC2 scores (D-SAU-Co). Several volatile compounds (3,5-octadien-2-one, 1-propanol, pentanal, 1-octen-3-ol, 1-penten-3-ol, acetic acid, 2-ethylfuran, 1-propene, allyl sulphide) are located in the right quadrants (upper and lower), which correspond to high positive charges in PC1, associated with D-SAU-Mo batch. On the other hand, there were a few volatile compounds allocated in the PC2: heptane, β-myrcene, and butanal in the left upper quadrant, and 2-pentyl-furan and 3-hydroxy-2-butanone in the left lower quadrant, related to D-SAU-Mu and D-SAU-Co, respectively. Thus, in comparison to C-SAU-Co and C-SAU-Mu, C-SAU-Mo are more related to typical volatiles compounds of fatty acid oxidation, such as 1-propanol, 1-octen-3-ol and pentanal [45], and to characteristic oxidation markers for PUFA oxidation, such as 1-penten-3-ol, 2-ethyl-furan and 3,5-octadien-2-one, which have been previously observed in mayonnaise and chicken nuggets enriched with fish oil [23,48], and are correlated with the strength of the oxidation process [48,50]. The low odor thresholds some of these volatile compounds, such as 1-octen-3-ol and 3,5-octadien-2-one (1 and 0.45 µg kg^−1^ of oil, respectively) [70,71], would lead to the perception of anomalous odor and/or flavor, which may have a negative impact in the products enriched with Mo. Nevertheless, in a previous study carried out with the D-SAU samples of the present work [28], no significant differences in acceptability were found among Co, Mo and Mu samples. So, the influence of the fish oil microcapsules addition on the profile of volatile compounds does not seem to be reflected in the consumer’s perception of D-SAU. 

Table 3 showed the individual volatile compounds of C-SAU as affected by the type of fish oil microcapsule added. α-thujene was the major volatile compound in all batches (around 6.07 AU × 10^6^), followed by pentanal (around 4.84 AU × 10^6^), β-thujene (around 3.52 AU × 10^6^), hexanal (around 3.5 AU × 10^6^), 1-octen-3-ol (around 3.40 AU × 10^6^), gamma-terpinene (around 3.20 AU × 10^6^) and heptanal (around 3.15 AU × 10^6^), and the rest individual volatile compounds showed less than 3 AU × 10^6^. In previous studies in cooked sausages, hexanal has been identified as the most abundant volatile compound, followed by heptanal, pentanal and volatiles compounds from the chemical families of alcohols (1-pentanol and 1-octen-3-ol) and terpenes (limonene, β-myrcene, and gamma-terpinene) [39,40,42], which is quite in agreement with the findings of the present study. However, in these previous works α-thujene and β-thujene were identified but with lower AU than in the present work. These compounds are associated with spicy flavor and have been found in a wide variety of medicinal herbs, essential oils, flavorings and spices such as nutmeg [72]; therefore, its abundance in the present study could be related to the addition of spices in the meat product formulation.

In C-SAU, the addition of Mo and Mu fish oil microcapsules significantly influenced the volatile compounds of most chemical families (Table 3), excluding terpenes and pyrazines, finding significant differences in 27 volatile compounds. Higher abundance was found in two volatile compounds in C-SAU-Mo (1-pentanol and 2-decenal) than in C-SAU-Co and C-SAU-Mu. A total of nine volatile compounds (hexane, heptane, decane, tridecane, 1-hexanol, phenyl ethyl alcohol, acetic acid, nonanoic acid, methyl propanoate and methyl propanoate) obtained higher AU in C-SAU-Mu than in C-SAU-Co and C-SAU-Mo. On the contrary, C-SAU-Mu showed lower abundance for 2-octene, nonane, 1-octen-3-ol, 2-methyl-propanal, 3-heptanone, 2-buthyl-furan and octanoic acid than C-SAU-Co and C-SAU-Mo. In comparison to the enriched batches, C-SAU-Co showed higher AU for six volatile compounds (pentanal, 2-heptanone, 2-methyl-furan, pentanoic acid, β-thujene and α-thujene) and lower for three (dodecane, 4-hexen-1-ol and 1-heptanol). A score plot of PCA of volatile compounds data from the C-SAU samples is shown in Figure 3b. The PC1 accounted for 54.78% of the total variance, and the PC2 comprised 34.19%. The score plot allowed a clear separation of the samples: those with high positive PC1 scores (C-SAU-Mu), those with high positive PC2 scores (C-SAU-Mo) and those with negative PC2 scores (C-SAU-Co). 2-decenal, 2-buthylfuran, 3-heptanone, 1-pentanol, 4-hexen-1-ol, 1-octen-3-ol, dodecane, nonane and methyl-propanoate were grouped and allocated in the upper quadrants (left and right), which correspond to the C-SAU-Mo batch, while 2-heptanone, pentanal, 2-methylpropanal, β-thujene, α-thujene, 2-methylfuran and hexane, heptane, decane, 1-hexanol, 1-heptanol, phenyl-ethyl-alcohol were in the lower left and right quadrants, related to C-SAU-Co and C-SAU-Mu, respectively. Thus, in C-SAU, Mo enriched batches also showed a close relation with volatile compounds from lipid oxidation, such us 2-decenal, a characteristic volatile compound of ω-3 PUFA oxidation, and in 1-pentanol, 4-hexen-1-ol and 1-octen-3-ol, described as typical lipid oxidation products [42,46]. 2-decenal has been related in previous studies to rancid odors in fish oil enriched mayonnaise [48] and fish oil microcapsules [20]. These volatile compounds have a fatty and fishy aroma [71] with a low odor threshold, around 10 µg kg^−1^ oil [73], which may be detrimental the acceptability of the meat products added with Mo. However, as occurred in D-SAU, similar scores in the acceptability analysis have been found by [28] in the three batches of C-SAU, the differences in the profile volatile compounds not being reflected by the sensory results.

It is worth noting that the differences found in the present study in the profile volatile compounds depend on the type of fish oil microcapsules added. Anyway, a major protection of the microencapsulated fish oil against lipid oxidation when Mu are added could be indicated. This can be ascribed to the different wall material in Mo (maltodextrine) and Mu (chitosan plus maltodextrine). Thus, the multilayer structured of chitosan-maltodextrine may protect the encapsulated material more effectively than the simple coating of maltodextrine. Indeed, chitosan improves the emulsion stability, increasing the electrostatic force and viscosity of the layers, and can also act as an antioxidant [63,64]. Moreover, a high oxidative stability has been found in microcapsules with chitosan [33]. So, although no marked effects on sensory or oxidation stability have been previously found [28], differences in the volatile compounds should be considered since they could release to unhealthy oxidized products [74]. This can be the case of furans, such us 2-ethylfuran, 2-butylfuran, 2-acetylfuran, 2-pentylfuran, 2-furfural and furfural alcohol, which have been found in different fish products [75,76], and have revealed toxicity in animals and humans [77,78]. In fact, 2-ethylfuran and 2-buthylfuran were closely related to the D-SAU-Mo and C-SAU-Mo samples in this study. Considering this aspect, more studies should be conducted in this sense, for evaluating the formation of contaminants in different omega-3 enriched meat products.

## 4. Conclusions

The type of fish oil microcapsules influences its profile of volatile compounds and that of the enriched meat products. The use of multilayered microcapsules with chitosan-maltodextrine walls may be more protective to the formation of lipid oxidation products, especially from omega-3 fatty acids, than microcapsules with a maltodextrine layer. Thus, the use of multilayered fish oil emulsions to elaborate omega-3 microcapsules for enriching meat products may be indicated.

## Figures and Tables

**Figure 1 foods-09-01683-f001:**
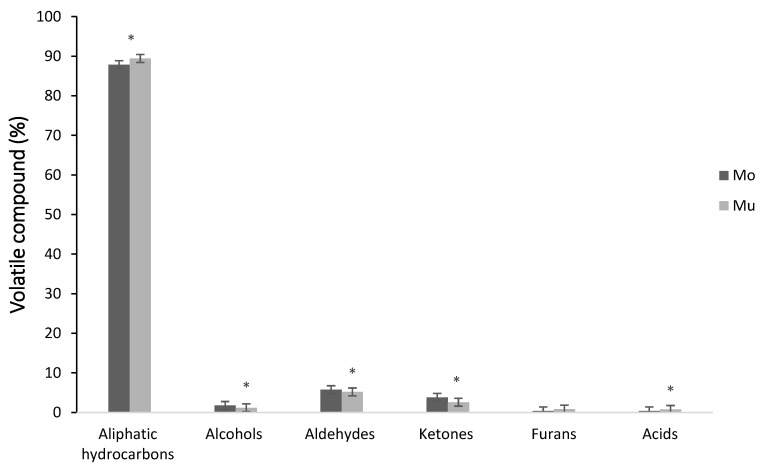
Volatile compounds of monolayered (Mo) and multilayered (Mu) fish oil microcapsules classified according to chemical families. Values are expressed as the average percentage of each family. *: significant differences (*p* < 0.05).

**Figure 2 foods-09-01683-f002:**
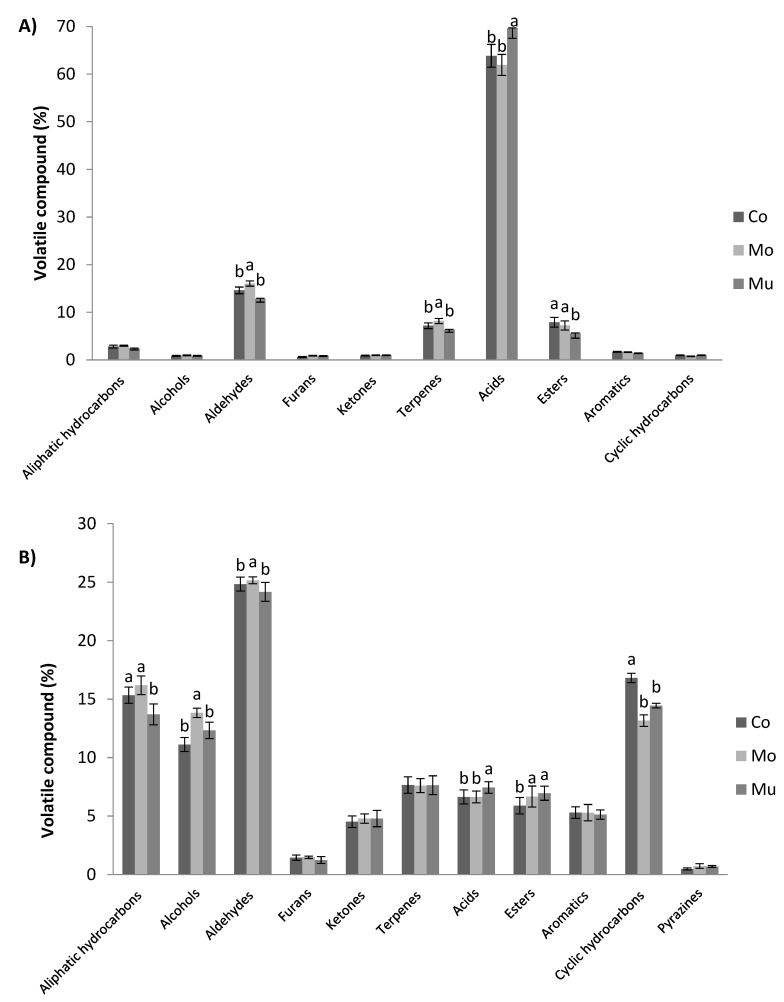
Volatile compounds of dry-cured (**A**) and cooked (**B**) sausages as affected by enrichment with omega-3 polyunsaturated fatty acids (ω-3 PUFA) (control: dark gray; enriched with multilayered fish oil microcapsules: medium gray; enriched monolayered fish oil microcapsules: light gray) classified according to chemical families. Values are expressed as the average percentage of each family. Bars with different letters (a, b) within the same formulation show significant differences (*p* < 0.05) due to enrichment effect (Co vs. Mo vs. Mu).

**Figure 3 foods-09-01683-f003:**
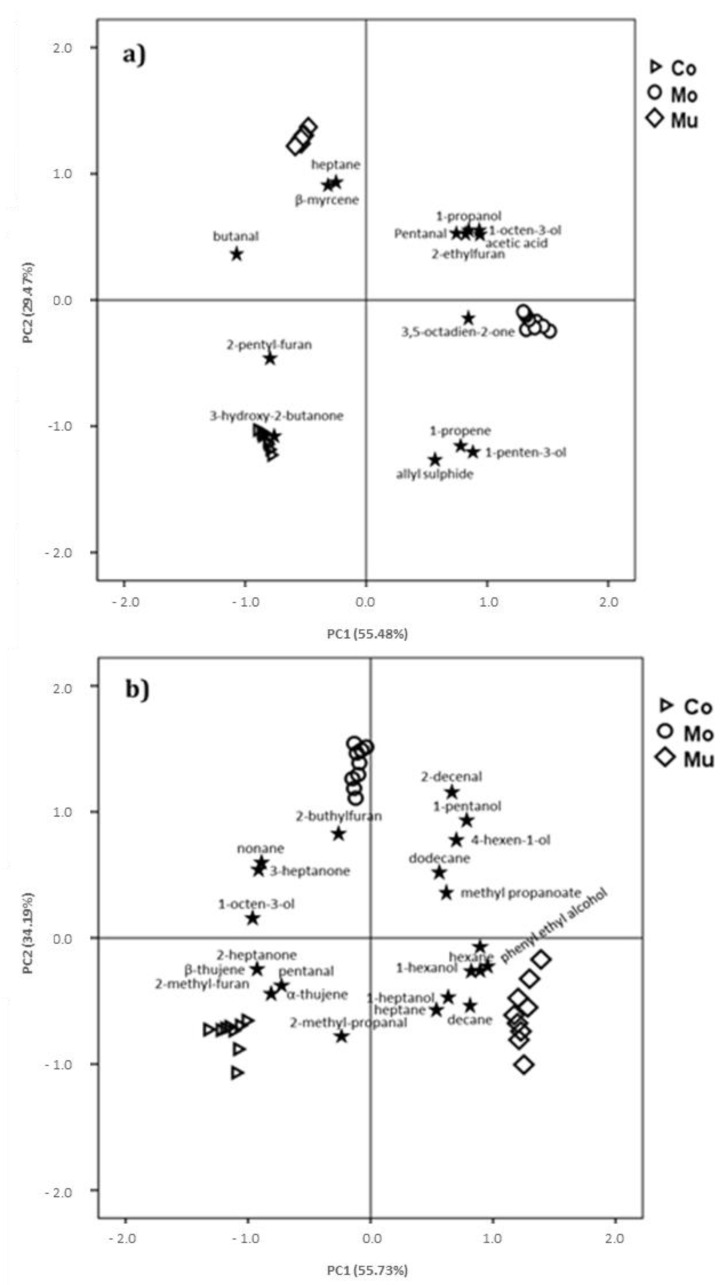
Principal component analysis (PCA) of the significant volatile compounds in dry-cured (**a**) and cooked (**b**) sausages. The plots represent, for the two first principal components, the loading of each volatile compound and the average scores of each one of batches. Control (▷); enriched with monolayered fish oil microcapsules (○); enriched with multilayered fish oil microcapsules (◊).

**Table 1 foods-09-01683-t001:** Volatile compounds from monolayered (Mo) and multilayered (Mu) fish oil microcapsules. Values are expressed as peak area × 10^6^.

LRI	ID	Compound	Mo	Mu	SEM	*p*
**Aliphatic hydrocarbons**						
504	A	Pentane	16.15	23.17	2.11	0.090
602	A	Hexane	91.47	83.26	2.47	0.091
692	A	1-heptene	0.03	n.d.	0.00	<0.001
703	A	Heptane	0.02	n.d.	0.00	0.004
799	A	Octane	0.05	0.07	0.01	0.002
901	A	Nonane	0.08	0.17	0.02	0.002
1003	A	Decane	0.02	n.d.	0.00	0.001
1099	A	Undecane	0.14	0.17	0.01	0.494
1200	A	Dodecane	0.02	0.08	0.01	0.013
1305	A	Tridecane	n.d.	0.04	0.01	0.001
**Alcohols**						
614	A	1-propanol	0.06	0.03	0.01	0.153
621	A	2-propanol	0.02	n.d.	0.00	0.007
687	A	1-penten-3-ol	0.29	0.08	0.05	0.026
784	A	2,3-butanediol	0.42	0.34	0.04	0.362
921	A	1,2,4-butanetriol	n.d.	0.55	0.12	<0.001
927	A	1-hexanol	0.08	0.01	0.02	<0.001
1015	A	4-hexen-1-ol	0.21	n.d.	0.05	0.008
1032	A	1-heptanol	0.03	n.d.	0.01	<0.001
1092	A	2-ethyl-1-hexanol	1.07	0.33	0.17	<0.001
1210	A	Phenylethyl alcohol	n.d.	0.07	0.02	<0.001
**Aldehydes**						
522	A	Propanal	0.27	0.15	0.03	0.048
587	A	2-methylpropanal	0.41	0.40	0.04	0.909
628	A	Butanal	0.45	0.29	0.04	0.055
730	A	Pentanal	1.49	0.26	0.29	0.003
777	A	2-pentenal	0.02	0.16	0.03	<0.001
854	A	Hexanal	2.83	1.94	0.20	<0.001
914	A	2-hexenal	0.07	n.d.	0.01	<0.001
936	A	Heptanal	0.66	0.24	0.10	0.014
1022	A	2-heptenal	n.d.	0.07	0.01	<0.001
1034	A	Octanal	1.42	0.49	0.23	0.016
1127	A	2-octenal	n.d.	0.48	0.12	0.009
1147	A	Nonanal	0.81	0.38	0.10	0.008
**Ketones**						
820	A	3-hydroxy-2-butanone	2.68	1.59	0.25	0.001
1070	B	1,4-cyclohexanedione	0.15	0.24	0.03	0.133
1140	A	2-nonanone	n.d.	1.10	0.25	<0.001
**Furans**						
722	A	2-ethylfuran	0.55	0.24	0.07	0.001
954	A	2-butyltetrahydrofuran	n.d.	0.02	n.d.	0.001
**Terpenes**						
1040	A	Limonene	0.02	0.01	n.d.	0.327
**Acids**						
1165	A	Heptanoic acid	n.d.	0.17	0.04	0.003
1168	A	Sorbic acid	n.d.	0.17	0.04	0.008

n.d. not detected; SEM. standard error; LRI. Linear retention index of the compounds eluted from the GC-MS; ID. Method of identification: A. mass spectrum and retention time similar to previous publication data; B. tentative identification by mass spectrum. *p* < 0.05 indicates significant differences between microcapsules.

**Table 2 foods-09-01683-t002:** Volatile compounds (arbitrary area units × 10^6^) on dry-cured sausages (D-SAU) as affected by enrichment with ω-3 PUFA (*p*) *.

LRI	ID	Compound	Co	Mo	Mu	*p*	SEM
**Aliphatic hydrocarbons**							
497	A	Pentane	0.49	0.44	0.65	0.409	0.15
599	A	Hexane	0.48	0.24	0.33	0.724	0.07
704	A	Heptane	0.41 ^b^	0.46 ^b^	0.92 ^a^	<0.001	0.08
761	A	1-propene	n.d. ^b^	0.43 ^a^	n.d. ^b^	<0.001	0.05
800	A	Octane	7.84	7.21	7.02	0.533	0.34
901	A	Nonane	0.52	0.39	0.63	0.057	0.04
997	A	Decane	0.89	1.01	0.92	0.319	0.07
1097	A	Undecane	1.95	1.81	1.89	0.756	0.11
**Alcohols**							
614	A	1-propanol	0.03 ^b^	0.12 ^a^	0.09 ^a^	0.043	0.02
687	A	1-penten-3-ol	n.d. ^b^	0.11 ^a^	n.d. ^b^	<0.001	0.26
820	A	1-pentanol	1.42	1.47	1.48	0.936	0.06
1024	A	1-heptanol	1.18	1.10	1.18	0.851	0.17
1031	A	1-octen-3-ol	2.58 ^c^	3.53 ^a^	3.03 ^b^	0.028	0.19
1195	A	4-terpineol	1.82	1.84	1.75	0.318	0.03
**Aldehydes**							
591	A	2-methylpropanal	0.31	0.42	0.28	0.172	0.03
618	A	Butanal	0.29 ^a^	0.10 ^b^	0.32 ^a^	0.002	0.05
667	A	2-methyl butanal	0.15	0.28	0.07	0.104	0.03
738	A	Pentanal	7.89 ^c^	9.54 ^a^	8.69 ^b^	0.027	0.33
862	A	Hexanal	92.26	93.64	101.51	0.176	6.08
942	A	Heptanal	8.12	7.27	7.35	0.433	0.62
1050	A	Octanal	0.49	0.55	0.53	0.525	0.02
1147	A	Nonanal	5.59	5.11	4.94	0.295	0.13
1322	A	2-decenal	0.87	0.84	0.88	0.594	0.01
1395	A	2,4-decadienal	0.39	0.42	0.37	0.577	0.04
**Ketones**							
749	A	2,3-pentanedione	0.47	0.45	0.46	0.456	0.03
778	A	3-hydroxy-2-butanone	0.49 ^a^	0.35 ^a^	0.15 ^b^	<0.001	0.04
933	A	2-heptanone	7.00	6.99	6.08	0.129	0.24
981	A	3-heptanone	3.34	3.76	4.95	0.330	0.40
1039	A	2-octanone	0.37	0.33	0.27	0.204	0.05
1063	A	3,5-octadien-2-one	n.d. ^b^	0.09 ^a^	n.d. ^b^	0.001	0.31
**Furans**							
722	A	2-ethyl-furan	n.d. ^b^	0.13 ^a^	0.07 ^a^	0.012	0.41
837	A	3-furaldehyde	0.39	0.26	0.22	0.053	0.02
1012	A	2-pentyl-furan	1.53 ^a^	1.09 ^b^	1.25 ^b^	0.004	0.06
**Terpenes**							
982	A	Sabinene	8.19	7.79	7.78	0.498	0.10
1003	B	β-myrcene	36.06 ^b^	36.84 ^b^	40.64 ^a^	0.023	0.70
1021	A	α-phellandrene	5.93	6.20	6.78	0.416	0.14
1037	A	d-limonene	3.55	3.23	3.82	0.168	0.26
1066	A	gamma-terpinene	2.63	2.27	2.41	0.122	0.10
1105	A	Terpene	1.87	1.87	1.60	0.280	0.11
1404	A	α-cubebene	1.23	1.26	1.68	0.282	0.09
**Acids**							
717	A	Acetic acid	458.69 ^b^	582.75 ^a^	422.62 ^b^	<0.001	43.49
895	A	Butanoic acid	30.15	26.94	27.08	0.094	1.26
986	A	Pentanoic acid	34.56	36.84	36.14	0.510	0.40
1362	A	Nonanoic acid	2.60	2.43	2.61	0.894	0.08
1472	A	Decanoid acid	3.05	2.58	2.61	0.409	0.17
**Esters**							
786	A	Methylpropyl acetate	0.27	0.28	0.26	0.992	0.02
853	A	Methyl hexanoate	65.15	51.94	57.08	0.345	2.55
**Aromatics**							
1018	A	Benzaldehyde	9.30	8.29	7.64	0.109	0.18
1190	A	4-methyl-phenol	0.53	0.59	0.60	0.517	0.02
1375	A	Eugenol	4.18	4.16	4.57	0.198	0.09
**Cyclic hydrocarbons**							
992	B	α-thujene	6.30	5.86	6.06	0.419	0.10
1495	A	Humulene	0.61	0.66	0.78	0.469	0.04
**Other**							
899	B	Allyl sulphide	2.41 ^a^	2.68 ^a^	1.90 ^b^	0.003	0.15

* Not enriched (Co) and enriched with monolayer (Mo) and multilayered fish oil microcapsules (Mu). Bars with different letters (a, b, c) within the same formulation show significant differences (*p* < 0.05) due to enrichment effect. n.d., not detected; SEM, standard error; LRI, linear retention index of the compounds eluted from the GC-MS; ID, method of identification: A, mass spectrum and retention time identical with an authentic standard; B, tentative identification by mass spectrum.

**Table 3 foods-09-01683-t003:** Volatile compounds (arbitrary area units × 10^6^) on cooked sausages (C-SAU) as affected by enrichment with fish oil microcapsules (*p*) *.

LRI	ID	Compound	Co	Mo	Mu	*p*	SEM
**Aliphatic hydrocarbons**							
499	A	Pentane	0.49	0.47	0.45	0.598	0.04
601	A	Hexane	0.44 ^b^	0.45 ^b^	0.78 ^a^	0.021	0.09
703	A	Heptane	1.49 ^b^	1.10 ^b^	2.03 ^a^	0.046	0.17
799	A	Octane	2.67	2.58	2.51	0.612	0.11
812	A	2-octene	2.05 ^a^	2.06 ^a^	1.67 ^b^	0.032	0.16
901	A	Nonane	1.89 ^a^	2.07 ^a^	1.03 ^b^	<0.001	0.13
1000	A	Decane	n.d. ^b^	n.d. ^b^	0.56 ^a^	<0.001	0.17
1101	A	Undecane	0.60	0.60	0.74	0.165	0.03
1200	A	Dodecane	0.40 ^b^	0.48 ^a^	0.54 ^a^	0.046	0.03
1296	A	Tridecane	n.d. ^c^	0.24 ^b^	0.46 ^a^	0.021	0.03
1402	A	Tetradecane	0.51	0.54	0.68	0.465	0.15
**Alcohols**							
615	A	1-propanol	0.13	0.11	0.14	0.561	0.10
825	A	1-pentanol	1.39 ^b^	1.73 ^a^	1.41 ^b^	0.028	0.04
923	A	1-hexanol	0.69 ^b^	0.80 ^b^	1.08 ^a^	0.039	0.06
927	A	4-hexen-1-ol	0.96 ^c^	1.90 ^b^	1.54 ^a^	0.026	0.12
1024	A	1-heptanol	0.66 ^b^	0.86^a^	0.91 ^a^	0.041	0.06
1031	A	1-octen-3-ol	3.84 ^a^	3.50 ^a^	2.84 ^b^	0.036	0.31
1088	A	2-ethyl-1-hexanol	0.44	0.44	0.57	0.134	0.14
1092	A	Phenyl ethyl alcohol	n.d. ^c^	0.37 ^b^	0.91 ^a^	0.011	0.06
**Aldehydes**							
593	A	2-methyl propanal	1.47 ^a^	1.21 ^b^	1.66 ^a^	0.048	0.11
667	A	2-methyl butanal	0.70	0.65	0.76	0.432	0.07
687	A	3-methyl butanal	2.04	2.19	2.25	0.686	0.53
738	A	Pentanal	6.06 ^a^	4.58 ^b^	3.89 ^b^	0.017	0.61
862	A	Hexanal	3.41	3.44	3.65	0.490	0.16
939	A	Heptanal	3.31	2.82	3.11	0.082	0.22
1047	A	Octanal	0.38	0.35	0.43	0.372	0.04
1114	A	2-octenal	0.08	0.13	0.18	0.142	0.05
1147	A	Nonanal	0.51	0.54	0.75	0.260	0.25
1328	A	2-decenal	0.77 ^b^	1.69 ^a^	0.89 ^b^	0.002	0.09
1390	A	2,4-decadienal	0.36	0.44	0.38	0.298	0.04
**Ketones**							
744	A	2,3-pentanedione	0.33	0.47	0.40	0.312	0.04
933	A	2-heptanone	0.84 ^a^	0.55 ^b^	0.38 ^b^	0.041	0.10
979	A	3-heptanone	2.27 ^a^	2.37 ^a^	1.05 ^b^	0.003	0.23
1039	A	2-octanone	1.11	1.35	1.21	0.087	0.09
**Furans**							
720	A	2-ethylfuran	0.38	0.26	0.21	0.446	0.08
908	A	2-butylfuran	0.37 ^a^	0.42 ^a^	0.19 ^b^	0.033	0.03
1008	A	2-pentyl-furan	0.79 ^a^	0.60 ^b^	0.45 ^c^	0.004	0.03
**Terpenes**							
1026	A	3-carene	1.56	1.38	1.42	0.311	0.12
1037	A	d-limonene	2.60	2.41	2.36	0.298	0.13
1066	A	gamma-terpinene	3.16	3.13	3.29	0.589	0.16
1097	A	Terpene	0.20	0.16	0.27	0.178	0.08
1491	A	Isocayophillene	0.26	0.24	0.26	0.589	0.03
**Acids**							
716	A	Acetic acid	n.d. ^b^	0.04 ^b^	0.22 ^a^	0.005	0.04
895	A	Butanoic acid	2.36	2.27	2.13	0.798	0.15
898	A	2-butenoic acid	0.25	0.29	0.26	0.636	0.03
986	A	Pentanoic acid	1.97 ^a^	1.06 ^b^	0.69 ^c^	<0.001	0.08
1273	A	Octanoic acid	0.42 ^a^	0.33 ^a^	n.d. ^b^	0.005	0.13
1366	A	Nonanoic acid	0.45 ^b^	0.43 ^b^	0.63 ^a^	0.032	0.04
1461	A	Decanoic acid	0.41	0.36	0.18	0.168	0.18
**Esters**							
656	A	Methyl propanoate	0.04 ^c^	0.22 ^b^	0.33 ^a^	<0.001	0.02
836	A	Ethyl butanoate	2.31	2.45	2.50	0.647	0.11
**Aromatics**							
1018	A	Benzaldehyde	1.59	1.94	1.67	0.109	0.19
1190	A	4-methyl-phenol	1.08	0.90	0.72	0.068	0.18
1305	B	Safrole	0.36	0.31	0.37	0.298	0.03
**Cyclic hydrocarbons**							
980	B	β-thujene	4.25 ^a^	2.76 ^c^	3.54 ^b^	0.029	0.41
991	B	α-thujene	7.95 ^a^	5.13 ^b^	5.13 ^b^	0.007	0.22
1422	B	cis-muurola-4(14),5-diene	0.20	0.19	0.19	0.469	0.02
1524	A	δ-cadinene	0.39	0.37	0.45	0.298	0.04
**Pyrazines**							
863	A	2-methylpyrazine	0.27	0.35	0.25	0.369	0.11
947	A	2,6-dimethylpyrazine	0.20	0.17	0.24	0.428	0.05

* Not enriched (Co) and enriched with monolayer (Mo) and multilayered fish oil microcapsules (Mu). Bars with different letters (a, b, c) within the same formulation show significant differences (*p* < 0.05) due to enrichment effect. n.d., not detected; SEM, standard error; LRI, linear retention index of the compounds eluted from the GC-MS; ID, method of identification: A, mass spectrum and retention time identical with an authentic standard; B, tentative identification by mass spectrum.

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
