# Peer review of "Effect of Omega-3 Microcapsules Addition on the Profile of Volatile Compounds in Enriched Dry-Cured and Cooked Sausages"

_foods, 2020, doi:10.3390/foods9111683_

Round 1

Reviewer 1 Report

Dear authors,   thank you for an interesting article with good content and well written. However, I have a few comments and suggestions:

Chapter 2.1: It would be good to describe better and in more detail (to explain) why the content of MO and MU in the prepared mixtures (2.75 %, 5.26 %, 3 and 5 g) is different. If I understand correctly, it is because the capsule shell has a different weight in both cases and in order to meet the condition of at least 40 mg EPS + DHA, then the addition must be higher in the case of MU than in the case of MO, is that so?

Chapter 2.2 .: Are the particles
formed by this preparation of the same or different size? What is the particle size? Can particle size affect the volatile compounds profiles?

Figure 1.: I would choose another type of graph, a bar graph, but only 2D. It would be good to add lines to the graph showing the variance of the percentage of content from different measurements - content variability (I mean something similar like error bars).
It would be good to make an enlarged section of the graph without a value for aliphatic hydrocarbons, otherwise the values ​​disappear.

lines 185 - 187: In the text I would not state the average values ​​of the content of the substance in MO and MU, it is misleading and the exact values ​​are given in the table.
The abbreviation AU should be explained.

picture 2 .: This picture is a bit confusing for me. It would be good not to mark the columns again with the letters a and b, it is confusing with the designation of the figure a and b and the description in the caption of the figure. The graphs should again have lines indicating the variance of individual values ​​within a given parameter. I do not see the letter c in the picture, although it is mentioned in the caption of the picture.

lines 278 et seq.: Same comment as for lines 185-187.

Overall, I think the statement: "around x %" doesn't sound very scientific. Personally, I would give either specific values ​​or a reference to a table with specific values, but that is just my opinion.

Table 2: Does the p value mean anything other than the pE value? Or is it the same parameter, just marked differently?

My suggestion: I would rather present the text and Table 3 first, together with the text and Table 2, and then summarize the statistics in a separate chapter, it would be clearer and better organized.

Figure 3: Perhaps an explanation could be added as to why there are 14 loadings in Figure 3a and 20 loadings in 3b?

Table 3: The tables should have the same graphic format.

Thank you, good job
Best regards

Author Response

The responses to the reviewer inquiries and the modifications included in the new manuscript are highlighted in yellow and listed below (the specific pages and lines correspond to the present version of the manuscript).

REVIEWER 1

Thanks for the comments.

Chapter 2.1: It would be good to describe better and in more detail (to explain) why the content of MO and MU in the prepared mixtures (2.75 %, 5.26 %, 3 and 5 g) is different. If I understand correctly, it is because the capsule shell has a different weight in both cases and in order to meet the condition of at least 40 mg EPS + DHA, then the addition must be higher in the case of MU than in the case of MO, is that so?

  • Considering this comment, the following sentence has been added “The different quantities of Mo and Mu added are due to the differences between Mo and Mu in the efficiency of fish oil encapsulation (87.39 and 56.43 %, respectively) and consequently, in the quantity of EPA and DHA (2.75 and 5.26 g EPA+DHA per 100 g of microcapsule, respectively).” (lines 81-84). The reason behind the different quantity of MO and MU added is the different quantity of EPA and DHA between these microcapsules.

Chapter 2.2 .: Are the particles formed by this preparation of the same or different size? What is the particle size? Can particle size affect the volatile compounds profiles?

  • Chapter 2.2. The particle size (mean diameter of Mo and Mu fish oil microcapsules) has been analyzed in a previous study using a laser light diffraction instrument, Mastersizer 300, not finding a marked effect on the mean particle diameters of Mo and Mu fish oil microcapsules. Quality characteristics of Mo and Mu emulsions and microcapsules have been previously analyzed and published (Solomando et al., 2019. DOI: 10.1111/jfpp.14290), which has been indicated in the present version of the manuscript (lines 109 and 110).

Figure 1.: I would choose another type of graph, a bar graph, but only 2D. It would be good to add lines to the graph showing the variance of the percentage of content from different measurements - content variability (I mean something similar like error bars).
It would be good to make an enlarged section of the graph without a value for aliphatic hydrocarbons, otherwise the values ​​disappear.

  • Figure 1. As the reviewer indicated, the type of graph has been modified to a 2D bar graph; in addition, error bars have been added to the graph that show the variance of the content percentage. Authors do no consider necessary to enlarge the section graph, since the percentage of all chemical families can be seen properly.

Lines 185 - 187: In the text I would not state the average values ​​of the content of the substance in MO and MU, it is misleading and the exact values ​​are given in the table.
The abbreviation AU should be explained

lines 278 et seq.: Same comment as for lines 185-187.

  • As the reviewer indicated, the average values related to the content of volatile compounds have been eliminated throughout the manuscript.
  • As the reviewer indicated, the abbreviation AU has been explained in line 155, chapter 2.4. Analysis of volatile compounds.

Picture 2 .: This picture is a bit confusing for me. It would be good not to mark the columns again with the letters a and b, it is confusing with the designation of the figure a and b and the description in the caption of the figure. The graphs should again have lines indicating the variance of individual values ​​within a given parameter. I do not see the letter c in the picture, although it is mentioned in the caption of the picture

  • As the reviewer indicated, the numbering of figures “a” and “b” has been replaced by a new numbering “1” and “2”, respectively, to avoid confusion with bars with different letters (a, b) within the same formulation due to the enrichment effect. As the reviewer indicated, error bars have been added. The letter c has been removed from the legend of Figure 2.

Overall, I think the statement: "around x %" doesn't sound very scientific. Personally, I would give either specific values ​​or a reference to a table with specific values, but that is just my opinion.

  • Authors are totally in agreement with the reviewer. This term has been eliminated throughout the manuscript.

Table 2: Does the p value mean anything other than the pE value? Or is it the same parameter, just marked differently?

  • Table 2. pE corresponds to the p value due to the enrichment with omega-3 PUFA microcapsules; to avoid confusion and according to the reviewer indications, it has been replaced by The same change has been done in Table 3.

My suggestion: I would rather present the text and Table 3 first, together with the text and Table 2, and then summarize the statistics in a separate chapter, it would be clearer and better organized.

  • Thanks for the suggestion. Authors have tried different organizations of the present paper, and, the current one seems to be the most accurate. The presentation of the results in one place and the statistics in a separate chapter may imply repetitions.

Figure 3: Perhaps an explanation could be added as to why there are 14 loadings in Figure 3a and 20 loadings in 3b?

  • Figure 3. The reason why there are different loadings in Figures 3a and 3b has been explained in the statistical analysis section (all volatile compounds that showed significant differences in the ANOVA analysis were included into a principal component analysis (PCA)) and also in the legend of Figure 3 "Principal component analysis (PCA) of the significant volatile compounds in dry-cured (a) and cooked (b) sausages". Thus, in figure 3.a there are 14 loads because there are 14 volatile compounds with significant differences between batches (table 2); and the same happens in figure 3.b, there are 20 loads that correspond to the significant differences shown in Table 3.

Table 3: The tables should have the same graphic format.

  • As the reviewer indicated, the graphic format of tables has been modified to have the same format in tables 1, 2 and 3.

Reviewer 2 Report

The paper titled “Effect of omega-3 microcapsules addition on the profile of volatile compounds in enriched meat products” deals within the scope of the Foods Journal, by investigating an interesting topic of research. However, some small improvements can be done. Please find below some remarks to help the revision of the manuscript.

Additional comments

Line 3: The expression “… in enriched meat products“ is too general. It would be better to use the term “…in dry-cured and cooked sausages” in the title.

Lines 54-58: The authors should provide a broader insight into current knowledge on oil microencapsulation and the issue of mono and multilayer films.

Line 103: Please provide more detailed recipe for sausage production (exact amounts of ingredients, equipment used) in order to ensure that the methods presented are repeatable by other researchers.

Line 111: Same as previous.

Lines 123-144: Please, provide the data on GC method validation.

Lines 162-181: In line 174 the authors stated: “Significant differences have been detected…”. Due to the way the results are presented in Figure 1, no conclusion can be drawn about the significance of the differences. Expressing the results in percentages can be misleading because the percentage of a single group of violate components for one microcapsule may be higher than for other, without actually being higher.  E.g., total sum of single compounds peak areas for aliphatic hydrocarbons is 107.98 for Mo and 106.96 for Mu, while the values in percentages are 87.89% and 90.43% for Mo and Mu, respectively. Expressing values in percentages may lead readers to conclude that Mo has more aliphatic hydrocarbons than Mu. I suggest that the mutual comparison of individual groups of violate components be made by summing the values (peak area) of the single components of particular group and then make a statistical analysis in order to detect differences in groups of violate compounds between the different types of microcapsules. Those results can be incorporate in the Table 2, along with the results of the individual volatile compounds.

The figure with percentages does not have to be deleted, but it must be made clear in the discussion that these percentages refer to single microcapsule (Mo or Mu) and should not be compared with respect to the type of microcapsules.

Lines 241-276: The previous comment can also be applied in the case of presenting the results of volatile compounds in dry-cured and cooked sausages.

Line 346 (Table 3): Remove the table shading.

Author Response

The responses to the reviewer inquiries and the modifications included in the new manuscript are highlighted in yellow and listed below (the specific pages and lines correspond to the present version of the manuscript).

REVIEWER 2

Thanks for the comments.

Line 3: The expression “… in enriched meat products “is too general. It would be better to use the term “…in dry-cured and cooked sausages” in the title.

  • Line 3. As the reviewer indicated, the tittle "Effect of omega-3 microcapsules addition on the profile of volatile compounds in enriched meat products” has been replaced by “Effect of omega-3 microcapsules addition on the profile of volatile compounds in dry-cured and cooked sausages”

Lines 54-58: The authors should provide a broader insight into current knowledge on oil microencapsulation and the issue of mono and multilayer films.

  • Since authors have recently published a paper review on these aspects, authors have considered more appropriate to reference it than to write a paragraph about it. However, information on Mo and Mu microcapsules have been added in the present version of the manuscript (lines 65-69).

Line 103: Please provide more detailed recipe for sausage production (exact amounts of ingredients, equipment used) in order to ensure that the methods presented are repeatable by other researchers.

Line 111: Same as previous

  • Chapter 2.3. Elaboration of meat products. As the reviewer indicated, more detailed information has been provided about the production of cooked and dry-cured sausages (exact amounts of ingredients and equipment used) (lines 112-131).

Lines 123-144: Please, provide the data on GC method validation.

  • Both SPME and GC-MS methods used in this work have been widely used for the identification and semi-quantification of volatile compounds from different food-matrices and microorganisms, including meat products: Food Microbiology. 92- 103556, pp. 1 - 10. Elesier, 2020; Food Research International. 100, pp. 691 - 697. 2017.

In fact, in the first version of the manuscript, the volatile analysis method has been referenced. Moreover, this methodology has been completed in the present version of the manuscript.

Lines 162-181: In line 174 the authors stated: “Significant differences have been detected…”. Due to the way the results are presented in Figure 1, no conclusion can be drawn about the significance of the differences. Expressing the results in percentages can be misleading because the percentage of a single group of violate components for one microcapsule may be higher than for other, without actually being higher.  E.g., total sum of single compounds peak areas for aliphatic hydrocarbons is 107.98 for Mo and 106.96 for Mu, while the values in percentages are 87.89% and 90.43% for Mo and Mu, respectively. Expressing values in percentages may lead readers to conclude that Mo has more aliphatic hydrocarbons than Mu. I suggest that the mutual comparison of individual groups of violate components be made by summing the values (peak area) of the single components of particular group and then make a statistical analysis in order to detect differences in groups of violate compounds between the different types of microcapsules. Those results can be incorporate in the Table 2, along with the results of the individual volatile compounds.

The figure with percentages does not have to be deleted, but it must be made clear in the discussion that these percentages refer to single microcapsule (Mo or Mu) and should not be compared with respect to the type of microcapsules.

Lines 241-276: The previous comment can also be applied in the case of presenting the results of volatile compounds in dry-cured and cooked sausages.

  • As the reviewer indicated, the results on the percentages of the samples of this study (shown in figure 1 and 2) are showing the general profile of the chemical families of volatile compounds. Comparison made on these results have been deleted, considering the reviewer comment. Results on individual volatile compounds (expressed in area units and shown in tables 1, 2 and 3) have been used to compare among samples.

  • Line 346 (Table 3): Remove the table shading.
  • As the reviewer indicated, the table shading has been removed.